# Mapping of Prion Structures in the Yeast Rnq1

**DOI:** 10.3390/ijms25063397

**Published:** 2024-03-17

**Authors:** Arthur A. Galliamov, Alena D. Malukhina, Vitaly V. Kushnirov

**Affiliations:** 1A.N. Bach Institute of Biochemistry, Federal Research Center “Fundamentals of Biotechnology” of the Russian Academy of Sciences, Moscow 119071, Russia; arturens96@gmail.com (A.A.G.);; 2Department of Biology, Moscow State University, Moscow 119991, Russia

**Keywords:** prion, amyloid, yeast, proteinase K, mass spectrometry, prion structure mapping

## Abstract

The Rnq1 protein is one of the best-studied yeast prions. It has a large potentially prionogenic C-terminal region of about 250 residues. However, a previous study indicated that only 40 C-terminal residues form a prion structure. Here, we mapped the actual and potential prion structures formed by Rnq1 and its variants truncated from the C-terminus in two [*RNQ*+] strains using partial proteinase K digestion. The location of these structures differed in most cases from previous predictions by several computer algorithms. Some aggregation patterns observed microscopically for the Rnq1 hybrid proteins differed significantly from those previously observed for Sup35 prion aggregates. The transfer of a prion from the full-sized Rnq1 to its truncated versions caused substantial alteration of prion structures. In contrast to the Sup35 and Swi1, the terminal prionogenic region of 72 residues was not able to efficiently co-aggregate with the full-sized Rnq1 prion. GFP fusion to the Rnq1 C-terminus blocked formation of the prion structure at the Rnq1 C-terminus. Thus, the Rnq1-GFP fusion mostly used in previous studies cannot be considered a faithful tool for studying Rnq1 prion properties.

## 1. Introduction

Amyloids are filamentous protein aggregates with a regular cross-beta-type structure. Amyloids are thought to be the cause of about forty incurable human diseases, although some amyloids play biologically important roles [1,2]. One of the human and animal amyloids, formed by the PrP protein, can be infectious and is called a prion because of this property. In yeast, many amyloids can propagate through cell generations, and in such cases, they are also called prions. Yeast prions are a very convenient and fruitful model for studying the basic properties of amyloids and prions. Since prion is defined as a “proteinaceous infectious particle”, some yeast phenomena of uncertain or non-amyloid nature also fit this definition [3,4,5], but we will not consider them in this work.

Currently, about nine yeast amyloid prions are known [6,7]. Two of the best-studied yeast prions are the Sup35 protein, which phenotypically manifests as a nonsense-suppressor [*PSI*+] determinant, and the Rnq1 protein, which manifests as a [*PIN*+] determinant that greatly enhances the de novo appearance of [*PSI*+] and other prions [8]. Since the [*PIN*+] phenotype can also be attributed to some prions other than Rnq1, we will use here the other, more specific name of this prion, [*RNQ*+].

In almost all yeast prion proteins, their prion-forming regions (PFRs) represent only a part of a protein, while the remaining part can serve a certain function. Thus, prion conversion may regulate this function. Rnq1 is unique in this regard because it has no clearly defined function, and it is possible that its only function is to facilitate the appearance of other prions in [*RNQ*+] cells. Most yeast PFRs are enriched in glutamine (Q) and asparagine (N) and are unstructured. In Rnq1, a PFR is usually associated with the large QN-rich C-terminal part of amino acid residues 153 to 405.

PFRs tend to be located in the terminal regions of the corresponding proteins. In five prions (Sup35, Ure2, Swi1, Mot3, Nup100), they are located at the N-terminus, in two (Rnq1, Lsb2) at the C-terminus, and only in Cyc8 PFR, they are internal. One more prion, Mod5, does not have a separate PFR and aggregates through its functional domain [9] and can, therefore, be excluded from this comparison.

Furthermore, the regions of PFRs closest to a terminus appear to be the most important for prion properties. The Swi1 protein has a large disordered QN-rich N-terminal region of 385 residues, but only the first 37 of them are required and sufficient for prion formation and propagation [10]. In Sup35, the prion structures can form within at least the first 150 residues, but only the first 70 or fewer of them are essential for prion formation and define the variant-specific prion phenotype [11,12]. The majority of Sup35 prion-eliminating mutations are located within its first 25 residues [13]. Consistent with these observations, protease mapping has revealed that Rnq1 has a single prion core formed by the last 40 residues [11], whereas the Rnq1 QN-rich PFR spans approximately 250 residues. In contrast, the deletion studies indicated that the sequences involved in Rnq1 prion propagation occupy at least 140 C-terminal residues [14,15].

It is generally believed that PFRs are portable, i.e., they retain their prion properties when fused to another protein. However, their context may be important. Fusion of the glutathione transferase to the N-terminus of Sup35 makes its PFR internal and completely blocks Sup35 prion properties [16]. Conversely, the placement of an internal PFR at a terminus can greatly enhance its prion properties. Sup35 has two PFRs at residues 2–70 and 90–120. If the second of these PFRs ends up at the C-terminus due to a nonsense mutation, the frequency of its spontaneous conversion to the prion state increases about six thousand-fold [17]. Studies of the [*RNQ*+] prion have mainly used Rnq1 hybrids with GFP or Sup35 fused to the Rnq1 C-terminus. In view of the above data, such instruments are likely to work incorrectly. However, their validity has never been tested.

In this work, we located, using partial proteinase K digestion, the actual and potential prion structures of Rnq1 in two [*RNQ*+] variants and show that they are not restricted to the C-terminal core of forty residues. While truncated Rnq1 proteins could readily acquire the prion fold from the native Rnq1, this caused significant structural rearrangements, even when deletions were small. We also show that the C-terminal GFP fusion to Rnq1 can alter its prion properties or block the prion transfer from Rnq1 to Rnq1-GFP. Thus, it is not fully correct to use the Rnq1-GFP fusion protein for studying Rnq1 prion properties.

## 2. Results

### 2.1. Rnq1 Constructs Used

Previously, we accidentally detected one of the Rnq1 prion cores (C-terminal Core 1, amino acid residues 366–405) while analyzing Sup35 prion preparations [11]. In these preparations, Sup35 was highly overproduced, while Rnq1 was produced at its modest natural level, suggesting that the Core 1 peptide has greatly increased visibility in the MALDI procedure.

To detect other Rnq1 prion cores, we created a set of gene constructs driving overproduction of the GFP-Rnq1 fusion protein and its variants with C-terminal Rnq1 deletions. We reasoned that either overproduction or the lack of Core 1 would allow us to observe other Rnq1 prion cores. GFP was placed at the N-terminus of Rnq1 to minimize its effect on prion formation, but we also studied the more common Rnq1-GFP fusion, which we took from the library of GFP fusions [18] (Figure 1). All constructs were placed on the multicopy pYes2 plasmid under control of the strong inducible *GAL1* promoter. The C-terminally deleted Rnq1 ended after residues 277, 355, 381, and 392. The latter two proteins also included two final residues of Rnq1, arginine and tyrosine, to potentially improve their stability. To test whether the C-terminal Core is sufficient for co-aggregation with a pre-existing Rnq1 prion, a pYes2 plasmid encoding GFP fusion to the last 72 residues of Rnq1 was created.

### 2.2. Aggregation Patterns of GFP-Labeled Rnq1 Proteins

The described plasmids were overproduced in two [*RNQ*+] variants of the 74-D694 strain. One is the variant originally residing in 74-D694 [19] and later designated as [*RNQ*+]-high [20]. The other variant is [*RNQ*+]-very high, characterized by Bradley et al. [20]. Both names refer to the frequency of appearance of another prion, [*PSI*+], seeded by these [*RNQ*+] variants. We will call these variants hereafter [*RNQ*+]-H and [*RNQ*+]-VH. The production of Rnq1 constructs was induced for 3 h and for 24 h, and then the aggregation patterns were photographed (Figure 2). Aggregation was observed in most but not all cases, and the aggregation patterns were different.

It should be noted that previous Sup35 studies revealed two aggregation patterns of the overproduced Sup35NM-GFP reporter protein. A large dot or several dots are formed when the reporter protein reproduces the fold of a pre-existing prion. When a prion or amyloid fold appears de novo, it usually forms elongated ring-like structures [21,22]. Rnq1 aggregates showed both the described and novel aggregation patterns. Rnq1-277 showed delayed formation of elongated ring-like structures, suggesting amyloid formation de novo due to the lack of pre-existing structures in the 1–277 region. Aggregation of Rnq72C was barely evident at 3 h, while at 24 h, it showed an unexpected novel pattern: a fibrous network at the cell periphery, often concentrated at or possibly originating from the bud neck. This looks more like a de novo appearance of an amyloid fold than reproduction of the existing one. Rnq72C was designed to contain the major C-terminal prion structure plus about 30 residues of a presumably unstructured, protease-sensitive region to avoid interference with the GFP globule. It was expected to mirror the Sup35(1-61)-GFP protein, which readily decorates all of the 23 Sup35 prion variants, forming numerous small dots [23] and faithfully reproducing at least three of the Sup35 variant-specific prion folds [12]. But Rnq72C turned out to behave very differently.

Rnq1-355, -381, and -392 formed single large dots at three hours, except that in [*PIN*+]-VH, Rnq1-381 aggregation was delayed, and all three proteins formed elongated protrusions from the main aggregate at 24 h. Gfp-Rnq1 showed another novel aggregation pattern. It initially formed a dot, but then a thin ring began to grow from the dot, eventually forming a network of thin fibers. This is clearly different from any of the Sup35-GFP aggregation patterns. Rnq1-GFP did not form elongated structures or bright dots. Its aggregates appeared to be a loose collection of many smaller particles.

### 2.3. Chromosomal Alterations of RNQ1

The C-terminal alterations analogous to the described pYes2 *RNQ1* constructs were introduced to the chromosomal *RNQ1* gene but with the native *RNQ1* promoter and without GFP. The *RNQ1* truncations were introduced to the [*RNQ*+]-H strain using the CRISPR/Cas9 technology. The *RNQ1*-GFP cassette was amplified from the library of GFP fusions [18], together with the associated *HIS3* marker, and used to replace *RNQ1* in [*RNQ*+]-H and -VH strains by homologous recombination with selection for histidine prototrophy.

The first aim was to test whether the C-terminal Rnq1 truncations or C-terminal GFP can impair [*RNQ*+] propagation. The chromosome integration approach should be and proved to be more stringent than the plasmid shuffle used in previous works [14,15,24] since it reduces the time during which full-sized Rnq1 and its altered variants coexist. The lysates of yeast cells with these alterations were analyzed by Western blotting. Each sample was loaded twice, with and without prior boiling. This allows us to assess the aggregation state of a protein since prion aggregates do not dissolve without boiling and do not enter a gel.

Electrophoretic analysis showed that C-terminal GFP caused prion loss in both [*RNQ*+] variants (Figure 3A,C). Another opportunity to test the effect of C-terminal GFP on [*RNQ*+] was to test the prion status of the *RNQ1*-GFP strain from the library of GFP-tagged yeast genes [18]. By analogy with the yeast genes deletion library [25], we expected the parent strain of this library, BY4741, to be [*RNQ*+], and this proved to be true. In contrast, the Rnq1-GFP protein in the *RNQ1*-GFP strain of this collection was soluble, i.e., non-prion (Figure 3C). Of note, the [*RNQ*+] variant of BY4741 is not known. It could be of common origin with [*RNQ*+]-H of 74-D694, but now it is difficult to establish. Thus, again, we observed a clear obstacle in the transition of prion state from Rnq1 to Rnq1-GFP. However, this obstacle can be overcome with prolonged coexistence of Rnq1 and Rnq1-GFP (Figure 3B).

Chromosomal truncations of *RNQ1* resulted in the loss of [*RNQ*+]-H in all cases except for Rnq1-392 (Figure 3A). However, the cause of the loss remained unclear because the levels of truncated proteins other than Rnq1-392 were reduced three to ten times, which could be one of the reasons for the prion loss. We did not try to reproduce this experiment with higher Rnq1 levels since this would be too laborious, while a similar experiment was performed earlier [14]. Our primary interest was to find out by genetic means which regions of Rnq1 have a prion structure. We assumed that the truncated Rnq1 proteins possessing these regions should co-aggregate in the presence of the native Rnq1 prion. The strains with truncated *RNQ1* that had lost a prion as a result of the truncation were crossed with the 21G-H67 strain that harbors [*RNQ*+]-H, and the cell lysates were examined by Western blotting. Co-aggregation was observed in all cases except for Rnq1-277 (Figure 3B). This indicates that the prion structures of the [*RNQ*+]-H are located downstream of residue 277. This agrees with the previous observations of Vitrenko et al., who observed that the Rnq(1-289)-GFP protein decorates the [*RNQ*+] prion, while Rnq(1-269)-GFP does not [14]. However, the deletion mapping data are in some contrast with the PK mapping data (Figure 4), which suggests that in the [*RNQ*+]-H, there is virtually no prion structure apart from the C-terminal core (residues 366–405). It is also worth noting that co-aggregation experiments usually use high levels of a GFP-fused tester protein combined with microscopy. Using lower levels is more correct, as at high levels, aggregation could result from heterologous seeding. For example, moderately overproduced Sup35 readily coaggregates with Rnq1 prion [26].

We also tested whether the chromosomally encoded truncated Rnq1 proteins can support prion propagation despite their reduced levels. For this, the respective GFP-Rnq1 proteins were overproduced, together with Sup35. Though [*RNQ*+] itself does not provide a selectable phenotype, [*RNQ*+] can be selected indirectly as a helper for the appearance of the Sup35 prion that can be selected by suppression of the *ade1-14* nonsense mutation. In this way, we were able to obtain the prion form of the Rnq1-355 and Rnq1-381 proteins. We also obtained the prion form of Rnq1-GFP in a similar way, although curiously, we could not do this by overproducing Rnq1-GFP but only by overproducing GFP-Rnq1.

### 2.4. Structural Maps of the Rnq1 Protein Constructs

The aggregates of Rnq1 constructs shown in Figure 1, except for Rnq72C, were isolated, and the location of their prion cores was determined using limited proteinase K (PK) proteolysis and mass-spectrometric identification of the PK-resistant peptides. The results are presented in Figure 4 and as individual graphs in Appendix A.

An important question in interpreting these results is how faithfully the overproduced protein reproduces the original prion fold, which propagates at modest natural Rnq1 levels, or how these folds are related. We expected three scenarios for the studied proteins: (A) faithful reproduction; (B) rearrangement; and (C) appearance de novo. The map for the GFP-Rnq1 in [*RNQ*+]-H is nearly identical to the data obtained without Rnq1 overproduction [11], which suggests faithful reproduction despite the formation of filamentous structures (Figure 2). GFP-Rnq1-277 and GFP-Rnq72C aggregated reluctantly through the formation of filamentous structures, which suggests, by analogy with Sup35 prion [21,22], amyloid formation de novo. The remaining proteins, excluding GFP-Rnq1-381 in [*RNQ*+]-VH, rapidly formed globular aggregates in which the original prion fold could be either preserved or rearranged.

#### 2.4.1. Structures of Full-Sized Rnq1 and Rnq1-GFP

In both [*RNQ*+] variants, the major prion structure in GFP-Rnq1 was the C-terminal core at residues 366–405. Almost no other structures were detected in [*RNQ*+]-H, while [*RNQ*+]-VH showed a more complex structure, with Cores 3 (285–323) and 4 (241–267, Figure 4 and Appendix A).

The C-terminal GFP fusion dramatically affected the prion structure. The Rnq1-GFP obtained in both [*RNQ*+] variants lacked Core 1 and Core 3 and became the main prion structure (Figure 4 and Appendix A). It is worth noting that the Core 3 structure in [*RNQ*+]-H was probably different from such cores in other cases. The GFP molecule, which was a part of all prion preparations, is highly resistant to PK, and usually, we observe only minor quantities of GFP-derived peptides. However, GFP peptides were major in the Rnq1-GFP from [*RNQ*+]-H. This could mean that the amyloid structure of Rnq1 had low resistance to PK. In any case, what is important about this amyloid is that no evidence of Core 1 was detected and that similar results were obtained for two [*RNQ*+]-H isolates obtained long ago from different laboratories, those of Y. Chernoff and S. Liebman.

**Figure 4 ijms-25-03397-f004:**
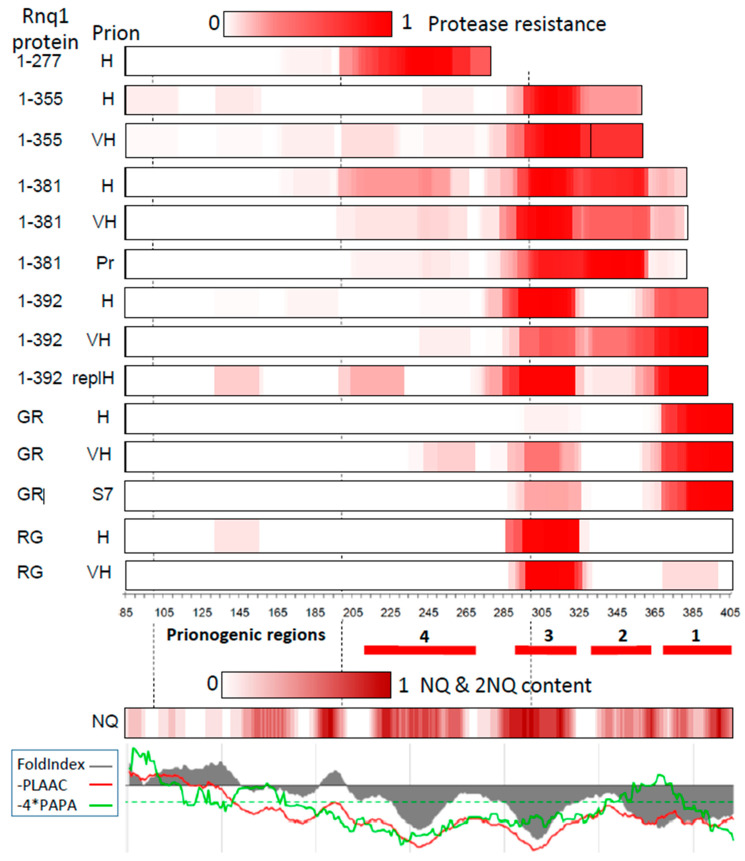
Protease-resistant cores of the GFP-Rnq1 and Rnq1-GFP proteins. Rnq1-derived proteins: 1-277 is GFP-Rnq1-277, 1-355 is GFP-Rnq1-355, etc.; GR is GFP-Rnq1; RG is Rnq1-GFP. These proteins were produced from pYes2 plasmid in prion strains: H is [*RNQ*+]-H, VH is [*RNQ*+]-VH, Pr: the named protein was encoded both in chromosome and pYes2 plasmid and its prion form was obtained de novo; S7: the amyloid was obtained de novo, being seeded by Sup35 prion of the “strong” [*PSI*+]-S7 variant [27]. Protease resistance is given according to Appendix A. The location of prionogenic regions 1 to 4 is indicated. The NQ scale shows the NQ content within a nine-residue window centered at a given residue. The lowest panel shows amyloidogenicity predictions by the PLAAC, PAPA, and FoldIndex algorithms generated at http://plaac.wi.mit.edu/ with default settings on 7 March 2024.

The Rnq1 prion is known to efficiently and rapidly seed Sup35 amyloid aggregation [26]. We attempted to perform the heterologous prion seeding in the opposite direction, hoping to obtain “naïve”, structurally preferable Rnq1 amyloid structures that are not selected for efficient seeding of the Sup35 prion. The aggregates of GFP-Rnq1 were obtained in the 74-D694 strain derivative lacking the *RNQ1* gene, where GFP-Rnq1 could be seeded with the Sup35 prion of the [*PSI*+]-S7 variant. The GFP-Rnq1 aggregation proceeded surprisingly inefficiently. The GFP-Rnq1 aggregates evident after three hours of overproduction proved to be liquid phase-separated droplets sensitive to 1,6-hexanediol. Such aggregates were observed in 39% or 82 of 211 of the examined untreated cells but only in 5.5% (15 of 275) of the cells treated with 1,6-hexanediol. This suggests that the GFP-Rnq1 aggregates were self-seeded rather than seeded by the Sup35 prion. These aggregates were allowed to mature and were collected after three days. Only about a quarter of them were insoluble in Sarcosyl. Thus, seeding of Rnq1 with the Sup35 prion was highly inefficient or did not occur at all. The obtained amyloid had prion structures in Regions 1 and 3, and by mapping, it was quite similar to GFP-Rnq1 from [*RNQ*+]-VH.

#### 2.4.2. Structures Observed in C-Terminally Deleted Rnq1 Variants

The GFP-Rnq1-392 protein lacks 11 C-terminal Rnq1 residues. This deletion caused significant alterations in the Rnq1 prion structure (Figure 4 and Appendix A). In the [*RNQ*+]-H, Core 3 became the major structure, while the decreased intensity of Core 1 could indicate that either this Core became much more protease-sensitive or it was present only in a small proportion of molecules. In the [*RNQ*+]-VH, the C-terminal Core significantly extended up to residue 329, which suggests a change of a prion fold compared to the full-sized Rnq1.

The GFP-Rnq1-381 protein lacks 22 C-terminal Rnq1 residues. This prevented the formation of the C-terminal core in [*RNQ*+]-H and induced prion structure formation in Regions 2 and 3 (residues 294–363) and 4 (198–257) (Figure 4 and Appendix A). These structures have no match in the original prion [*RNQ*+]-H and thus apparently result from rearrangement of the original structure.

The GFP-Rnq1-355 protein formed structures in Regions 2 and 3, but this amyloid was apparently heterogeneous and included two separate structures, one ending at the C-terminus (Tyr-355) and occupying the Region 2 and 3 and the other occupying the Region 3 only (Appendix A). This conclusion is based on the observation that the former structure is represented by peptides with random starting points but all ending at residue 355, while the latter structure was represented by peptides with a clear tendency for their starting point (residues 285, 294) and ending point (311, 323, 328). Thus, rearrangement of the initial prion resulted in two new different structures. Some PK-resistant peptides were also detected in regions 85–112, 132–155, and 241–267, but we do not have sufficient evidence that they belong to the amyloid structure.

The GFP-Rnq1-277 protein formed amyloid de novo and could, therefore, only indicate the localization of amyloidogenic regions. Its PK-resistant peptides were located in the region 198–277 (Figure 4 and Appendix A). The combination of their start and end points suggests that they represent two or more independent populations, as would be expected for a de novo amyloid. Importantly, the peptides from this region with similar preferred start and end points were also observed in some of the other Rnq1 preparations studied.

## 3. Discussion

### 3.1. On the Protease Mapping Procedure

Yeast prion structure usually includes both structured and unstructured regions. The former store and reproduce the specific fold of a prion, but the latter are equally important. They are recognized by the chaperones of Hsp40 and Hsp70 families that result in prion multiplication through fragmentation performed by Hsp104 [28,29]. This is critical for prion propagation, but also, this defines to a large extent the specific properties of a prion: the size of prion particles and the proportion between aggregated and soluble prionogenic protein. PK digestion allows for mapping of the structured and unstructured regions, thus providing information that defines the properties of a prion.

### 3.2. Rnq1 Prion Structures

The mapping data revealed four regions where prion structures form or can be formed (Figure 4). It should be noted that, most likely, each of these regions can form more than one prion structure. Sometimes, it is evident by some difference in the location of these structures within a particular region, but sometimes, there may be a difference in structure without a difference in location.

The most notable of the Rnq1 prion structures is the C-terminal Core 1 (residues 366–405). It was the major structure in both studied [*RNQ*+] variants, as well as in the GFP-Rnq1 preparation seeded by the Sup35 prion. It appears likely that any Rnq1 prion variant should have the C-terminal Core, similar to the N-terminal Core of Sup35, universally present in all 26 of its mapped prion isolates [11]. The formation of Core 1 could be inhibited by small C-terminal deletions (Rnq1-392, Rnq1-381) or C-terminal GFP attachment (Rnq1-GFP in [*RNQ*+]-VH, Figure 4).

The next important prionogenic region is Region 3, where the structure appeared in all cases except for GFP-Rnq1 in [*RNQ*+]-H. The structures in Region 2 appeared only in truncated Rnq1. These structures were very different. In GFP-Rnq1-392/[*RNQ*+]-VH, the Region 2 structure was an extension from Region 1; in Rnq1-381 prion, there was a single structure occupying Regions 2 and 3; Rnq1-381 in [*RNQ*+]-H and [*RNQ*+]-VH formed a mix of the latter structure plus separate structures in Regions 2 and 3; Rnq1-355 in [*RNQ*+]-H formed a mix of two structures: the C-terminal one, including Regions 2 and 3, and a structure in Region 3. The presence of multiple structures in Rnq1-355 is shown in Appendix A. The structural heterogeneity of these preparations is not surprising. Although we have observed that the prion fold readily transitions from the full-size Rnq1 to its truncated versions (Figure 2), this fold appears to change and may transform into different structures in different populations of cells.

The Rnq1-277 protein lacks Regions 1 to 3 and did not co-aggregate with Rnq1 in [*RNQ*+]-H. However, overproduced GFP-Rnq1-277 formed Sarcosyl-insoluble aggregates, which probably represent amyloid structures formed de novo. These structures occupied the C-terminal part of the protein, the region 198–277. Curiously, most of the PK-resistant peptides in this preparation belonged to four groups ending at residues 247, 257, 267, and 277 (Appendix A), although there are no obvious repeats in this region to match this ten-residue increase. Structures located within this region were also observed in some other preparations (Figure 4).

Finally, PK-resistant structures were also observed in some preparations at residues 85–112, 132–155, and 166–193 (Figure 4). Two of these locations belong to the N-terminal region regarded as non-amyloidogenic. However, it would be difficult to prove the amyloid nature of these structures. Some PK-resistant peptides could belong to non-amyloid structures. For example, these are the GFP fragments we often see in mass spectra.

### 3.3. General Considerations

This work shows that different experimental approaches can lead to fundamentally different results. As a good example, the Rnq1 prion transfers to the Rnq1-GFP protein in plasmid shuffle mode but does not transfer when chromosomal *RNQ1* is exchanged for *RNQ1*-GFP. Obviously, both results are correct, but they tell us different parts of the story. The plasmid shuffle experiment shows that the Rnq1-GFP protein can acquire and propagate the prion state of Rnq1. The chromosomal *RNQ1* replacement shows that the prion transition to Rnq1-GFP encounters significant resistance, most likely related to the need for a large or small rearrangement of the prion fold.

Apart from the above difference, the aggregation properties of GFP-Rnq1 and Rnq1-GFP differed in several assays (Figure 2 and Figure 4). This indicates that Rnq1-GFP is a poor tool for studying the Rnq1 prion. However, the majority of publications used Rnq1-GFP or Rnq1-Sup35C, and only one paper used a fluorescent protein correctly attached to the Rnq1 N-terminus [30]. While in many cases the use of Rnq1-GFP did not dramatically affect the results, in at least one case it did; likely due to prion structural rearrangements, Rnq1-GFP is particularly unsuitable for studying the [*RNQ*+] variant-specific differences performed in [24].

The ability of prions to seed the appearance of other prions [8] is a medically important phenomenon, as human amyloids also often cross-seed each other [31]. While pathological amyloids usually appear in old age, some amyloids are functional and appear early in life. Can they promote the appearance of pathological aggregates? At present, it is not known what defines the efficiency of amyloid cross-seeding. A simple assumption would be that it is some kind of similarity of amyloidogenic regions. However, this is not true because, as we observed, seeding can be asymmetric. Seeding of Rnq1 by the Sup35 prion was very inefficient, in contrast to seeding in the opposite direction.

An interesting question is how the mapped structures (Figure 4 and Appendix A) correlate with computer predictions. An earlier work [24] used predictions of Rnq1 amyloidogenic regions by five algorithms: (1) Zyggregator (http://wwwvendruscolo.ch.cam.ac.uk/zyggregator.php); (2) PASTA (http://biocomp.bio.unipd.it/pasta/); (3) TANGO (http://tango.crg.es/); (4) Aggrescan (http://bioinf.uab.es/aggrescan/); (5) WALTZ (http://waltz.switchlab.org/). Among the predictions were mainly hydrophobic peptides, including the repeating sequence SFSALASMASSY. In our results, prion structures mainly correlated with high QN content, while hydrophobic sequences separated Cores 2, 3 and 4. Thus, the correlation with our data was more negative than positive. Just one algorithm, Zyggregator, predicted the peptide SFNFSGNFS (386–394) located within the major structure of the Rnq1 prions, C-terminal Core 1. However, it could not predict that the actual structure is much larger, starts earlier, and extends to the C-terminus.

A much better prediction was made by the algorithm PLAAC [32,33] (Figure 4, bottom panel), which is not surprising since it was trained on four yeast prions, including Rnq1, though without a good knowledge of the precise location of amyloid structures. Even more definite predictions were made by FoldIndex [34], the algorithm that predicts the folding propensity of a sequence irrespective of its prion potential. Finally, the PAPA algorithm [35] made a good prediction for the C-terminal Region 1 but could not distinguish other detected prion/amyloid structures.

### 3.4. Problems to Be Solved

The Rnq1 prion turned out to be more complicated than we had expected, based on our previous experience with the Sup35 prion. The prion transfer from Rnq1 to its truncated versions caused significant changes in the prion fold. In some cases, aggregation of the truncated Rnq1 proteins occurred with a time delay. The GFP-Rnq72C protein was reluctant to decorate the Rnq1 prion particles, in contrast to analogous Sup35-GFP constructs that readily decorated the Sup35 prion. Some microscopic aggregation patterns were unusual. GFP-Rnq1 in the [*RNQ*+] cells formed a filamentous network that was never observed for Sup35. A possible explanation is that overproduced Rnq1 depletes chaperones more efficiently than Sup35, and the Rnq1 prion fibers quickly cease to be fragmented. In the same cells, the Rnq1-GFP aggregates resembled loosely bound collections of smaller particles. Further experiments are needed to understand these observations. For example, a more detailed study of the nature and dynamics of the prion transition between Rnq1 and its deletion variants.

### 3.5. Concluding Remarks

The following results of this work appear to be the most important:The actual and potential prion structures of Rnq1 were mapped, and their location better correlates with high QN content than with predictions of some computer algorithms.Prion structures significantly rearrange when transferring from native Rnq1 to its truncated versions or to Rnq1-GFP. This implies that in the cases when such a transfer is successful, one most likely obtains a different prion fold.The reluctance of GFP-Rnq72C to aggregate was unexpected, since the “mirror” Sup35 protein Sup35(1-61)-GFP readily decorates different Sup35 prion variants. Thus, some important details of how prion co-aggregation occurs are not well understood and require further investigation.The Rnq1 prion appears to differ from Swi1 and Sup35 in that the key Rnq1 prion structure is not restricted to a single terminal prion core.While the Rnq1 prion readily induces the prion state of Sup35, the opposite is not true: the Sup35 prion seeds Rnq1 very inefficiently, if at all.Prion transition from Rnq1 to Rnq1-GFP does not occur when the Rnq1 protein is changed rapidly through chromosomal replacement of the *RNQ1* gene but occurs when the change proceeds during many cellular generations through plasmid shuffle. The C-terminal Rnq1 prion core disappears in Rnq1-GFP overproduced in both studied [*RNQ*+] prion variants. The microscopic aggregation patterns of Rnq1-GFP differ significantly from that of GFP-Rnq1. Thus, Rnq1-GFP does not appear to be an adequate instrument for studying Rnq1 prion properties. In general, GFP or other tags should not be fused to the same side of a protein as its PFD.

Overall, due to the presence of several prionogenic regions, the behavior of Rnq1 prions and especially their interaction with truncated Rnq1 proteins is very complicated and far from being fully understood, which should be the subject of further studies.

## 4. Materials and Methods

### 4.1. Yeast Strains and Media

Yeast strain 74-D694 (*MATa ade1-14 ura3-52 leu2-3,112 his3-Δ200 trp1-289* [*RNQ*+]-high) [19] and its derivative with a different [*RNQ*+]-very high prion variant [20] were mainly used. The 74-D694 derivative harboring “strong” [*PSI*+] variant OT56 [36], also known as S7 [11], was further modified by deletion of the *RNQ1* and *PRB1* genes [27] and kindly provided by A. Dergalev. The strain 21G-H67 (*MATalfa ura3 leu1 lys1 ade2-1 SUQ5 kar1 cyhR* [*RNQ*+]-High), obtained by M. Ter-Avanesyan previously, was used as a mating partner.

Synthetic complete (SC) media contained 6.7 g/L Yeast Nitrogen Base without amino acids (Difco), 20 g/L glucose or galactose, and required amino acids.

### 4.2. Plasmids and DNA Manipulations

For the production of various Rnq1 fusions with GFP protein, the respective gene constructs were placed on the pYes2 (Thermo Fisher Scientific, Waltham, USA) multicopy *URA3* vector under control of a strong inducible *GAL1* promoter. To construct the pYes2-sGFP-Rnq1 plasmid, pYes2-sGFP-Sup35(1-112) [17] was cut with SfoI (EheI) & XbaI, which removes the *SUP35* fragment to replace it with *RNQ1*. The *RNQ1* coding sequence was amplified by PCR using primers sGFP-Rnq-Df and Yes-Rnq-Rf (all primers are listed in Table 1). The plasmid and *RNQ1* fragment were joined using a ligase-less Quick-fusion cloning kit (Vazyme, Nanjing, China) to create pYes2-sGFP-Rnq1 plasmid. The same technique was used for construction of all pYes2-based plasmids. For the C-terminally truncated Rnq1 constructs, specific reverse primers were used instead of Yes-Rnq-Rf primer; for the RNQ72C construct, the sGfp-Rnq72cDf primer was used instead of sGFP-Rnq-Df. To create pYes2-RNQ1-GFP, a DNA fragment encoding the Rnq1-GFP was amplified from the genome of the *RNQ1*-GFP strain from the library of GFP-tagged genes [18] using pYes-Rnq1-Df and pYes-GFP65-Rf primers. This fragment was joined to pYes2 by PvuII and XbaI sites.

To create chromosomal *RNQ1* truncations analogous to the plasmids with indexes 277, 355, 381, and 392, *RNQ1* was amplified with Rnq1-Din and a respective reverse primer was used for plasmid construction, Yes-RnqXXX-Rf. The *RNQ1* 3′ terminator region was amplified by YesXba-Rnq3′D and Rnq1-R1. The latter fragment was joined to each of the former fragments by PCR, thus creating DNA repair cassettes for CRISPR/Cas9 chromosomal editing. Thus, these cassettes included 20 nucleotides of pYes2 terminator region between the truncated *RNQ1* and *RNQ1* terminator, which should not affect the protein production. Chromosomal Cas9 editing was performed with the help of the pWS172 multicopy *HIS3* plasmid [37], to which two variants of the *RNQ1* targeting sequence were ligated. Two pairs of targeting oligonucleotides formed two DNA duplexes with sticky ends, each of which was ligated into the pWS172 Esp3I site. One of the obtained plasmids, pWS172-Rnq1-CrC, targeted Cas9 to the sequence corresponding to Rnq1 residues 369–375 and was used to make *RNQ1*-277, -355, and -381 replacements. The other plasmid, pWS172-Rnq1-Cr2, targeted Cas9 to the sequence starting from the *RNQ1* last codon and was used to make the *RNQ1*-392 replacement.

To perform CRISPR/Cas9 editing, yeast cells were transformed with the appropriate pWS172 plasmid and DNA repair cassette. Transformation was performed as described by Gietz and Woods (2002).

### 4.3. Genetic Manipulations

To test the ability of the truncated Rnq1 proteins to co-aggregate with native Rnq1 prion, the 74-D694 strains, where [*RNQ*+] was lost due to truncation of the *RNQ1* gene or its replacement for *RNQ1*-GFP, were crossed to the 21G-H67 [*RNQ*+]-high on the SC medium lacking leucine.

The prion form of the altered Rnq1 proteins was obtained as described [27]. Briefly, the 74-D694 [*rnq*-] strains with altered chromosomal *RNQ1* were transformed with pYes2 plasmids with the same *RNQ1* alteration and with multicopy *LEU2* plasmid Yeplac181 carrying the *SUP35* gene with its functional C domain replaced with GFP. The production of altered Rnq1 was induced for two days, and then the cells were plated to SC medium lacking adenine. This selects the nonsense suppressor [*PSI*+] cells, which are usually also [*RNQ*+].

### 4.4. Microscopy

Fluorescence microscopy was performed by the agar pad technique [38] using an Axioskop 40 fluorescence microscope (Zeiss, Oberkochen, Germany). To confirm the aggregation of Rnq1-GFP into liquid phase-separated droplets, cells were treated with 10% 1,6-hexanediol for 30 min prior to microscopy, according to [39].

### 4.5. Determination of the Rnq1 Prion State

Yeast cell lysates were separated by SDS-PAGE and subjected to Western blotting with an anti-Rnq1 antibody [40]. Each lysate was loaded twice, with and without prior boiling. Prions do not solubilize without boiling and so do not enter a gel. Thus, there should be a difference between the two loadings when a protein is a prion and no difference when a protein is in a non-prion state. Blots were decorated with rabbit polyclonal anti-Rnq1 antibody and goat anti-rabbit peroxidase-conjugated secondary antibody (Pierce #31460) and developed with the Pierce Supersignal West Dura kit.

### 4.6. Isolation of Prion Aggregates

The Rnq1 prion and amyloid aggregates were isolated as described previously [11], with some modifications. The cells with multicopy plasmids encoding the Rnq1 variants under control of the *GAL1* promoter were grown in 150 mL of non-inducing SC + glucose medium to OD600 = 2.5. Then, 150 mL of inducing SC + galactose medium was added, and the culture was grown overnight. Cells were collected into 50 mL Falcon tubes (about 3 g of wet pellet, optimally) and lysed by vigorous shaking for 6 min with 2 cm^3^ of glass beads and 1.5 mL TBS (30 mM Tris-HCl, pH 7.6; 150 mM NaCl) with 3 mM PMSF, 1 mM dithiothreitol and 10 µg/mL of RNase A and 10 µg/mL DNase. Sucrose gradients were made in 15 mL Falcon tubes, including 1 mL each of 60, 50, 40, 30, and 20% sucrose with TBS and 1 mM PMSF. The lysates, including cell debris, were diluted to 7 mL, loaded on top of sucrose gradients, and spun at 4300 RPM, 4 °C for 40 min. The fractions containing GFP were transferred to one or two 2.3 mL Eppendorf tubes without significant effort to avoid cell debris. The material was spun for 3 min at 16,000× *g* to remove sucrose. The pellets were resuspended in 1.5 mL of TBS with 1 mM PMSF. Sarcosyl was added to 5%, and the mix was sonicated for 30 (6 × 5) s at 50% power in a VCX130 sonicator with a 2 mm tip (Sonics and Materials, Newtown, CT, USA). Lysate was centrifugated at 20,000× *g*, 4 °C for 3 min. The supernatant was loaded into a 3.5 mL open-top tube (Beckman Coulter cat. 349622) on top of a sucrose gradient made of 150 μL each of 60, 50, 40, 30, and 20% sucrose with TBS and 0.5% Sarcosyl. Tubes were spun in an MLS-50 bucket rotor (Beckman Coulter, Brea, CA, USA) at 268,000× *g* for 3 h. GFP-Rnq1 aggregates were visually identified by GFP fluorescence in the 60% sucrose fraction and collected by pipette.

### 4.7. PK Digestion, Mass Spectrometry, and Data Analysis

Prion preparations (200 μg/mL) were digested by PK (25 μg/mL) in 20 µL for 1 h at room temperature. PK was inactivated by adding 1 µL of 100 mM PMSF, and peptides were precipitated by addition of 16 µL of acetone and incubation on ice for 10 min (0.5 mL Eppendorf tubes were used). Peptides were collected by centrifugation at 16,000× *g* for 1 min, washed with 80% acetone, dissolved in 10 μL of water, denatured by boiling for 3 min, and analyzed by MALDI-TOF/TOF mass spectrometer UltrafleXtreme (Bruker Daltonics, Billerica, MA, USA).

The spectra were taken in two modes: reflection and linear. Peptides were identified by MS-MS and/or as groups of related peaks in the reflection spectra, but the amounts of PK-resistant peptides were taken from the linear-mode spectra. To graphically represent the PK-resistant structures, the PK resistance index R was calculated for every residue as a sum of mass-spectral peak areas of all peptides, which include this residue. The R values were normalized against their maximum for each preparation to fit the range of 0 to 100% (Appendix A). These data were also converted into heat maps using Microsoft Excel 2013 (Figure 4). To depict the NQ content, the number of asparagine and glutamine residues was calculated for each nine-residue sequence window centered at a particular residue and divided by 9.

## Figures and Tables

**Figure 1 ijms-25-03397-f001:**
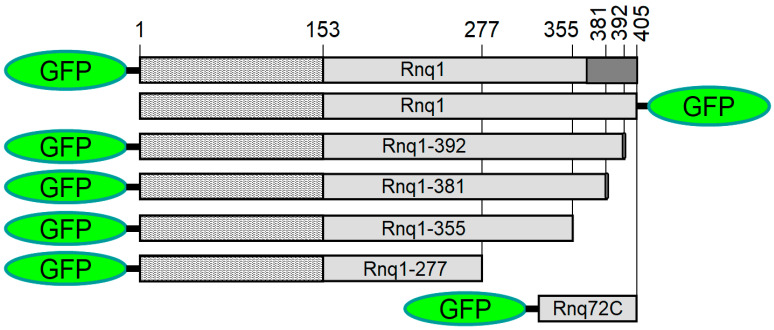
The hybrid Rnq1 proteins used to observe aggregation patterns and map prion structures. The genes encoding these proteins were located on a multicopy pYes2 plasmid under control of the *GAL1* promoter. Amino acid residues 1–152: presumable non-prionogenic region, residues 153–405: QN-rich prionogenic region. Dark grey box: the prion core at residues 366–405 observed earlier [11].

**Figure 2 ijms-25-03397-f002:**
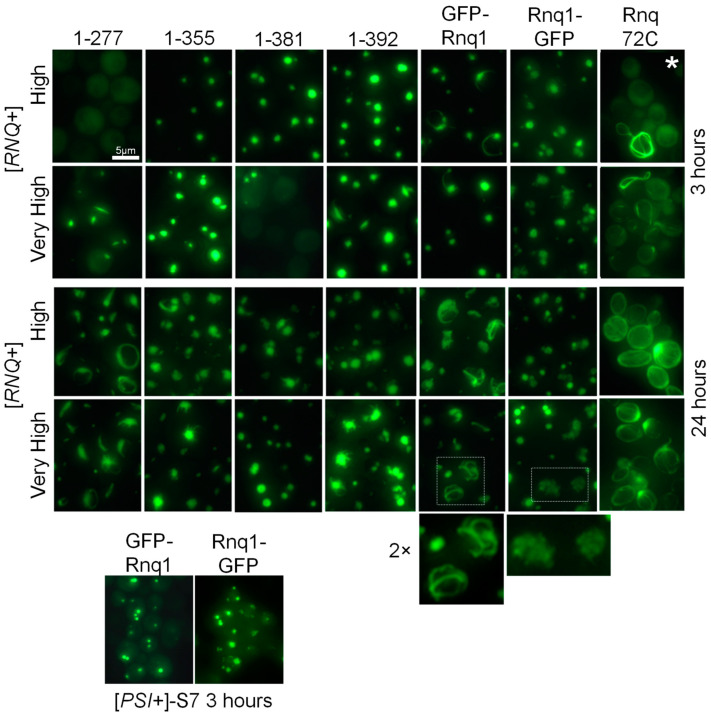
Aggregation patterns of GFP-labeled Rnq1 protein constructs. [*RNQ*+] variants are indicated on the left, and Rnq1 constructs are shown on top. Two-fold magnification of the boxed areas is shown. The [*PSI*+]-S7 strain is *∆rnq1*. *Ring-like aggregates in this sample were observed in only about 2% of cells.

**Figure 3 ijms-25-03397-f003:**
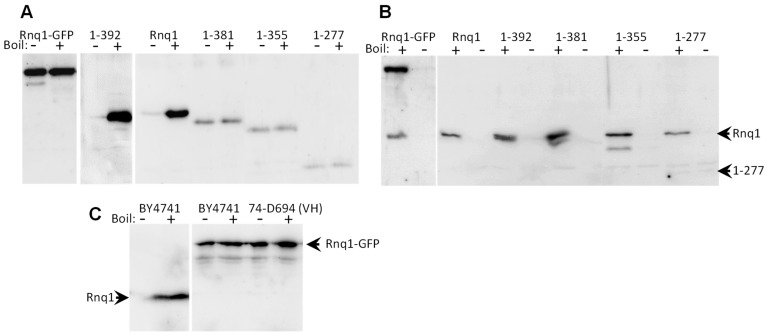
Replacement of the Rnq1 protein by its truncated variants or Rnq1-GFP can cause loss of the prion. Cell lysates were loaded for electrophoresis with or without boiling and Western blotted with immunostaining for Rnq1. Prion aggregates do not enter polyacrylamide gel but dissolve when boiled. (**A**) Lysates of the descendants of the 74-D694 [*RNQ*+]-H strain with the *RNQ1* gene were replaced with the indicated *RNQ1* truncated versions or *RNQ1*-GFP. Only Rnq1-392 acquired the prion form. (**B**) The strains from panel A were crossed to the 21G-H67 [*RNQ*+]-H: all Rnq1 variants co-aggregated, except for Rnq1-277. (**C**) The BY4741 strain is [*RNQ*+], but its *RNQ1*-GFP descendant lacks prion, the same as 74-D694 [*RNQ*+]-VH with *RNQ1* replaced with *RNQ1*-GFP.

**Table 1 ijms-25-03397-t001:** Oligonucleotide primers used.

Primer	Sequence	Description
sGFP-Rnq-Df	gatgaactgtacaaaggc**caaagatctgaaATGgatacgga**	sGFP to *RNQ1* junction
Yes-Rnq-Rf	*tacatgatgcggccctctaga* **atcatcgtTCAgtagcggt**	*RNQ1* to pYes2 junction
Yes-Rnq277x-Rf	*catgatgcggccctctaga*TTA**gtaagattgagccatggag**	*RNQ1*-277 to pYes2 junction
Yes-Rnq355x-Rf	*catgatgcggccctctaga*TTA**atactcattagcctgttgctg**	*RNQ1*-355 to pYes2 junction
Yes-Rnq381-Rf	*catgatgcggccctctaga* **TCAgtagcgattggagttctggtttccgcc**	*RNQ1*-381 to pYes2 junction
Yes-Rnq392-Rf	*catgatgcggccctctaga* **TCAgtagcggttgccagaaaaattgaagga**	*RNQ1*-392 to pYes2 junction
sGfp-Rnq72cDf	ctgtacaaaggccaaagatc**ctacctgggcaataactcc**	sGFP to *RNQ72C* junction
Rnq1-CrC-D	gact**tacggaagaccgcaatacgg**	CRISPR spacer for pWS172 plasmid
Rnq1-CrC-R	aaac**ccgtattgcggtcttccgta**	CRISPR spacer for pWS172
Rnq1-CR2-D	gact**actgaatcatcgttcagtag**	CRISPR spacer for pWS172
Rnq1-CR2-R	aaac**ctactgaacgatgattcagt**	CRISPR spacer for pWS172
Yes-Rnq1-Df	*cggatcggactactagcag* **caaagatctgaaATGgatacg**	pYes2 to *RNQ1* junction
Yes-GFP65-Rf	*atgatgcggccctctaga* cgcgccctatttgtatagtt	pYes2 to GFP junction
Rnq1-Din	**gggagccaaagtatgggtg**	*RNQ1* inside ORF
Rnq1-R1	**gggcatcctgcagagataca**	*RNQ1* terminator
YesXba-Rnq3′D	*tctagagggccgcatcatg* **cgatgattcagttcgccttc**	PCR of *RNQ1* 3′ region

Sequences related to pYes2 are italicized, those related to GFP are underlined, and those related to *RNQ1* are bold. *RNQ1* start and stop codons are in uppercase. Primers ending with D or Df are on the coding strand, those ending with R or Rf are reverse primers. The first four nucleotides of the CRISPR oligonucleotides form sticky ends for ligation to the Esp3I site of pWS172 [37].

## Data Availability

Data is contained within the article and Appendix A.

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
