# Peer review of "Mapping of Prion Structures in the Yeast Rnq1"

_ijms, 2024, doi:10.3390/ijms25063397_

Round 1

Reviewer 1 Report

Comments and Suggestions for Authors

This manuscript examines the amyloid cores of prions formed by Rnq1. By utilizing both N- and C-terminal GFP fusions, and detailed protease mapping, the results nicely add to previous work examining Rnq1 truncations. It provides interesting insights into the ability of prion cores to change upon transfer between different proteins, and acts as a cautionary tale about the effects that GFP can have on prion cores. I have just a few criticisms, which should be largely addressable with text edits.

1.       The authors reference previous work by Vitrenko et al examining Rnq1 truncations. It might be useful to provide a little more detail about these previous results, to give more context for the current experiments.

2.       The authors should indicate the source of the antibody used for western blots.

3.       On page 6 (line 216), it appears that there is a missing reference (the word “Ref” appears as a placeholder in the text).

4.       In Figure 4, the authors indicate four “prionogenic” regions. They should describe in more detail how they defined these four regions. In particular, it is not obvious from the protease mapping that Region 2 is any more prionogenic than the region from about 165-195: both are protected in a subset of structures but are never the sole protected region.

5.       In lines 249-251, the authors state that after one day, GFP-Rnq1 had only formed liquid droplets, but there is no indication of how they determined these droplets to be liquid.

6.       In the last paragraph of the results, the authors state that GFP-Rnq1-277 seems to form multiple distinct structures. Could this be tested, by isolating single colonies to see if individual isolates show just one of these structures?

7.       The authors indicate that QN content is a better predictor of prionogenic regions than computer algorithms, but surprisingly, they don’t use any of the computer predictions that were developed for yeast prion proteins (PLAAC, PAPA, PrionW, etc.). The PLAAC server incorporates both PLAAC and PAPA. While PLAAC identifies the entire region from about 150-405, it appears that there is reasonable correlation between the strongest PLAAC peaks and the prionogenic regions; PAPA’s strongest signals roughly correspond to Regions 4 and 1. pWALTZ identifies the entire segment from 201-405, so identifies all four prionogenic regions, but doesn’t identify boundaries between them. A little more discussion of prion-specific algorithms might be interesting.

Author Response

We thank this Reviewer for his/her notes that helped to noticeably improve the manuscript. We managed to address all concerns and provide the answers below.

In addition, some important alterations were made to the mapping results (Figures 4 and S1). The protease resistance profiles were added for the Rnq1-355 and -381 proteins in the [RNQ+]-VH strain. These two were missing, since only their [RNQ+]-H counterparts were present. A more significant change is that we replaced the protease resistance profile for the Rnq1-GFP in the [RNQ+]-H. We were not fully satisfied with the quality of the submitted profile and remade it for two [RNQ+]-H strains. (We have two such strains, obtained from Y. Chernoff and S. Liebman. They existed independently for a long time and, as a result, one of them even proved to be a diploid). In contrast to the old Rnq1-GFPprofile, new ones lack the C-terminal core (residues 366-405) similarly to the Rnq1-GFP in [RNQ+]-VH strain. We are confident in the new data and consider the old data to be an error. We also removed the PK map for the Rnq1-GFP in the Rnq1-GFP prion, which resulted from another error, and we hope that no more errors are left. The corrected data strengthen our conclusion that C-terminal GFP alters the Rnq1 prion fold and now we can conclude that it blocks formation of the C-terminal Rnq1 core.

Reviewer 1

This manuscript examines the amyloid cores of prions formed by Rnq1. By utilizing both N- and C-terminal GFP fusions, and detailed protease mapping, the results nicely add to previous work examining Rnq1 truncations. It provides interesting insights into the ability of prion cores to change upon transfer between different proteins, and acts as a cautionary tale about the effects that GFP can have on prion cores. I have just a few criticisms, which should be largely addressable with text edits.

  1. The authors reference previous work by Vitrenko et al examining Rnq1 truncations. It might be useful to provide a little more detail about these previous results, to give more context for the current experiments.

Answer: We have added some details to this episode: "This agrees with previous observations of Vitrenko et al., who observed that the Rnq(1-289)-GFP protein decorates the [RNQ+] prion, while Rnq(1-269)-GFP does not [14]."

  1. The authors should indicate the source of the antibody used for western blots.

Answer: The description of antibodies was added: "Blots were decorated with rabbit polyclonal anti-Rnq1 antibody and goat anti-rabbit peroxidase conjugated secondary antibody (Pierce #31460), and developed with Pierce Supersignal West Dura kit."

  1. On page 6 (line 216), it appears that there is a missing reference (the word “Ref” appears as a placeholder in the text).

Answer: Thank you for noting. The already used references #21, #22 were inserted here (Zhou et al., 2001, Tyedmers et al., 2010).

  1. In Figure 4, the authors indicate four “prionogenic” regions. They should describe in more detail how they defined these four regions. In particular, it is not obvious from the protease mapping that Region 2 is any more prionogenic than the region from about 165-195: both are protected in a subset of structures but are never the sole protected region.

Answer: the data in favor of the Region 2 being amyloidogenic are that it formed a joint structure with either Region 3 (in Rnq1-381 preparations) or Region 1 (in GFP-Rnq1-392/[RNQ+]-VH). But probably the best evidence gives the newly added PK-resistance profile for GFP-Rnq1-355 in [RNQ+]-VH, in which the major (though still not the only) structure is located in the Region 2 (Figure S1E).

  1. In lines 249-251, the authors state that after one day, GFP-Rnq1 had only formed liquid droplets, but there is no indication of how they determined these droplets to be liquid.

Answer: We have added the data and the method for this. The text was added to the section 2.4.1: "The GFP-Rnq1 aggregates evident after three hours of overproduction proved to be liquid phase-separated droplets sensitive to 1,6-hexanediol. Such aggregates were observed in 39%, or 82 of 211 of the examined untreated cells, but only in 5.5% (15 of 275) of the cells treated with 1,6-hexanediol."

  1. In the last paragraph of the results, the authors state that GFP-Rnq1-277 seems to form multiple distinct structures. Could this be tested, by isolating single colonies to see if individual isolates show just one of these structures?

Answer: This could probably be tested, but it would be difficult and time-consuming, while the value of such a result is not entirely clear. One would have to induce protein production and obtain amyloid, then take several individual cells and grow them for about 40 generations to about 2-3 g of wet cell mass without turning off expression and hoping that the cells do not develop any adaptations to avoid the burden of overproduction.

  1. The authors indicate that QN content is a better predictor of prionogenic regions than computer algorithms, but surprisingly, they don’t use any of the computer predictions that were developed for yeast prion proteins (PLAAC, PAPA, PrionW, etc.). The PLAAC server incorporates both PLAAC and PAPA. While PLAAC identifies the entire region from about 150-405, it appears that there is reasonable correlation between the strongest PLAAC peaks and the prionogenic regions; PAPA’s strongest signals roughly correspond to Regions 4 and 1. pWALTZ identifies the entire segment from 201-405, so identifies all four prionogenic regions, but doesn’t identify boundaries between them. A little more discussion of prion-specific algorithms might be interesting.

Answer: Thank you for this note, which helped us to understand that some algorithms are good enough. The PLAAC made sufficiently good predictions, and surprisingly, FoldIndex was even better, though it is not dedicated to prions. PAPA was so-so. We could not cope with PrionW which constantly reported some incomprehensible errors. And WALTZ predicted two small peptides. Apparently, working with such programs requires a good knowledge of optimal settings. Anyway, two favorites are FoldIndex and PLAAC. We have added the PLAAC result to Figure 4 and the following paragraph to the text:

"A much better prediction was made by the algorithm PLAAC [32,33] (Figure 4, bottom panel), which is not surprising, since it was trained on four yeast prions including Rnq1, though without a good knowledge of the precise location of amyloid structures. Even more clear predictions were made by FoldIndex [34], the algorithm which predicts the folding propensity of a sequence irrespectively of its prion potential. Finally, the PAPA algorithm [35] made a good prediction for the C-terminal Region 1, but could not distinguish other detected prion/amyloid structures."

Reviewer 2 Report

Comments and Suggestions for Authors

In this work, the authors used partial proteinase K digestion to locate the actual and potential prion structures of Rnq1 in two [RNQ+] variants and showed that they are not restricted to the C-terminal core of forty residues. They also showed that the C-terminal GFP fusion to Rnq1 can alter its prion properties or block the prion transfer from Rnq1 to Rnq1-GFP. Thus, it is not fully correct to use Rnq1-GFP fusion protein for studying Rnq1 prion properties.

In summary, this manuscript was very well-written, very comprehensive, and the experiments were very carefully designed and performed. All the backend data was also provided in detail. The significance and novelty were clearly stated in the "conclusion" section. After all, I think this paper can be published with the current form. Only some minor points:

For Figure 2, it is blurry, could the authors increase the DPI or quality of the image? Can the authors also provide the "scale bar" of the images?

For Figure 3 and 4, the content and image are also blurry, could the authors provide with a high quality data in the supplementary file?

The behavior of Rnq1 prions and especially their interaction with truncated Rnq1 proteins is very complicated and far from being fully understood. What are the next steps the authors could try to understand this problem better? Please list this in the conclusion as well.

Author Response

We thank this Reviewer for his/her notes that helped to noticeably improve the manuscript. We managed to address all concerns and provide the answers below.

In addition, some important alterations were made to the mapping results (Figures 4 and S1). The protease resistance profiles were added for the Rnq1-355 and -381 proteins in the [RNQ+]-VH strain. These two were missing, since only their [RNQ+]-H counterparts were present. A more significant change is that we replaced the protease resistance profile for the Rnq1-GFP in the [RNQ+]-H. We were not fully satisfied with the quality of the submitted profile and remade it for two [RNQ+]-H strains. (We have two such strains, obtained from Y. Chernoff and S. Liebman. They existed independently for a long time and, as a result, one of them proved to be a diploid). In contrast to the old Rnq1-GFPprofile, new ones lack the C-terminal core (residues 366-405) similarly to the Rnq1-GFP in [RNQ+]-VH strain. We are confident in the new data and consider the old data to be an error. We also removed the PK map for the Rnq1-GFP in the Rnq1-GFP prion, which resulted from another error, and we hope that no more errors are left. The corrected data strengthen our conclusion that C-terminal GFP alters the Rnq1 prion fold and now we can conclude that it blocks formation of the C-terminal Rnq1 core.

Reviewer 2:

In this work, the authors used partial proteinase K digestion to locate the actual and potential prion structures of Rnq1 in two [RNQ+] variants and showed that they are not restricted to the C-terminal core of forty residues. They also showed that the C-terminal GFP fusion to Rnq1 can alter its prion properties or block the prion transfer from Rnq1 to Rnq1-GFP. Thus, it is not fully correct to use Rnq1-GFP fusion protein for studying Rnq1 prion properties.

In summary, this manuscript was very well-written, very comprehensive, and the experiments were very carefully designed and performed. All the backend data was also provided in detail. The significance and novelty were clearly stated in the "conclusion" section. After all, I think this paper can be published with the current form. Only some minor points:

For Figure 2, it is blurry, could the authors increase the DPI or quality of the image? Can the authors also provide the "scale bar" of the images?

For Figure 3 and 4, the content and image are also blurry, could the authors provide with a high quality data in the supplementary file?

Answer: The scale bar (5μm) was present in the first image of the Figure 2 (1-277/high).

Blurriness: In the reviewed version we provided raster images, but with this version we provide vector graphics, which should solve the problem. The images are also provided as original PowerPoint and PDF files.

However, the former images were not of poor quality, at least as I see them in our original Word file. The blurriness could be an effect of software compatibility with other programs.

The behavior of Rnq1 prions and especially their interaction with truncated Rnq1 proteins is very complicated and far from being fully understood. What are the next steps the authors could try to understand this problem better? Please list this in the conclusion as well.

Answer: We were quite puzzled by some of our observations and at present we do not have particularly good ideas about the direction of further research. All we can suggest now is a more detailed study of the nature and dynamics of the prion transition between Rnq1 and its deletion variants. The latter phrase was added at the end of the section "3.4. Problems to be solved".